# Robust Assessment of Macromolecular Fraction (MMF) in Muscle with Differing Fat Fraction Using Ultrashort Echo Time (UTE) Magnetization Transfer Modeling with Measured T1

**DOI:** 10.3390/diagnostics13050876

**Published:** 2023-02-24

**Authors:** Saeed Jerban, Yajun Ma, Qingbo Tang, Eddie Fu, Nikolaus Szeverenyi, Hyungseok Jang, Christine B. Chung, Jiang Du, Eric Y. Chang

**Affiliations:** 1Department of Radiology, University of California, La Jolla, San Diego, CA 92093, USA; 2Radiology Service, Veterans Affairs San Diego Healthcare System, La Jolla, San Diego, CA 92161, USA; 3Department of Orthopedic Surgery, University of California, La Jolla, San Diego, CA 92093, USA

**Keywords:** muscle, MRI, UTE, myotendinous junction, fat infiltration

## Abstract

Magnetic resonance imaging (MRI) is widely regarded as the most comprehensive imaging modality to assess skeletal muscle quality and quantity. Magnetization transfer (MT) imaging can be used to estimate the fraction of water and macromolecular proton pools, with the latter including the myofibrillar proteins and collagen, which are related to the muscle quality and its ability to generate force. MT modeling combined with ultrashort echo time (UTE-MT modeling) may improve the evaluation of the myotendinous junction and regions with fibrotic tissues in the skeletal muscles, which possess short T2 values and higher bound-water concentration. The fat present in muscle has always been a source of concern in macromolecular fraction (MMF) calculation. This study aimed to investigate the impact of fat fraction (FF) on the estimated MMF in bovine skeletal muscle phantoms embedded in pure fat. MMF was calculated for several regions of interest (ROIs) with differing FFs using UTE-MT modeling with and without T1 measurement and B1 correction. Calculated MMF using measured T1 showed a robust trend, particularly with a negligible error (<3%) for FF < 20%. Around 5% MMF reduction occurred for FF > 30%. However, MMF estimation using a constant T1 was robust only for regions with FF < 10%. The MTR and T1 values were also robust for only FF < 10%. This study highlights the potential of the UTE-MT modeling with accurate T1 measurement for robust muscle assessment while remaining insensitive to fat infiltration up to moderate levels.

## 1. Introduction

Magnetic resonance imaging (MRI) is widely regarded as the most comprehensive imaging modality to assess skeletal muscle quality and quantity. Muscle inflammation, fat infiltration, fibrosis, and atrophy are the most common qualitative observations in MR images of muscle injuries and pathologies. Conventional MRI techniques that are routinely performed for qualitative muscle assessments include T1- and T2-weighted [1], short-tau-inversion-recovery [2,3], and diffusion-weighted imaging [4]. However, the standard morphological MRI-based muscle assessments are subjective and may lack reproducibility. Quantitative MRI techniques, including T1 and T2 relaxation times [5,6,7,8,9,10,11], magnetization transfer (MT)-related measures [10,11,12,13,14,15,16,17,18], and diffusion tensor (DT) indices [4,19,20,21], as well as fat and water separation [22,23,24,25,26,27,28,29,30], may provide more sensitive, objective, and reproducible measures of the muscle microstructure and composition.

Among the above-mentioned quantitative MRI techniques, MT can provide information on proton pools, such as water and macromolecules, using MT ratio or size estimation of the pools (e.g., macromolecular fraction, MMF) [12,13]. In muscle, the dominant macromolecules include the myofibrillar proteins and collagen, and a measurement of this pool could be potentially sensitive to the muscle quality and its ability to generate force [31,32]. Using MT techniques, a high-power saturation radio frequency (RF) pulse is used with a predefined frequency different from the water protons’ resonance frequency to saturate mainly protons in macromolecules in the skeletal muscle fibers. After the saturation pulse application, the saturated magnetization transfers from the macromolecules to the water protons (bound and free water) that can be detected by MRI as a signal reduction. MT imaging combined with ultrashort echo time (UTE) MRI has been recently introduced as a technique to incorporate the saturations experienced by both free and bound water in different biological tissues [33,34]. UTE-MT modeling may particularly improve the evaluation of the myotendinous junction and regions with fibrotic tissues in the skeletal muscles, which possess short T2 values and higher bound-water concentration. Detecting the applied saturation in such regions of muscle is challenging with conventional MRI sequences that utilize echo times (TEs) greater than a few milliseconds [35]. Accurate MT modeling requires B1 correction and T1 compensation [36]. Remarkably, UTE-MT modeling has been shown to be insensitive to tissue orientation and to provide a robust assessment of musculoskeletal (MSK) tissues regardless of their orientation within the MRI scanner [37].

Fat protons, in theory, should not participate in the MT phenomena because the lipid proton spin group is essentially isolated from the water protons and from the protons on large macromolecules [38]. However, the fat present in muscle has always been a source of concern in MMF calculation and a few studies have reported significant underestimation of MMF for muscles with considerable fat infiltration [16]. This could pose a problem for some applications, such as the rotator cuff musculature in which pathologic muscles can contain up to 30% fat [39]. Robust estimation of MMF in muscle regardless of the macroscopic fat infiltration will be a useful tool for muscle assessment decoupled from the fat infiltration. 

The purpose of this study was to investigate the impact of fat fraction (FF) on the estimated MMF in bovine skeletal muscle phantoms embedded in pure fat (lard). MMF was calculated for several regions of interest (ROIs) with differing FFs using UTE-MT modeling with and without T1 measurement and B1 correction.

## 2. Material and Methods

### 2.1. Phantom Preparation

Fresh cuts of lean bovine chuck muscles and pure pork fat (lard) were purchased from a local grocery store. The muscle cuts were visually examined to select a lean portion, avoiding the inclusion of obvious interfascicular fat in the prepared muscle specimen. Two muscle sections were placed in a cylindrical plastic container (10 cm in diameter and 20 cm length, approximately). Lard was melted at 60 °C and then cooled to near room temperature. Next, melted lard was poured into the container with the two muscle sections at the bottom. Lard and muscle sections filled up to the middle of the container. The muscle-fat phantom was kept at room temperature until the lard reached its stable solid state. The container was topped with tap water to ensure performing MR imaging at the water peak frequency.

### 2.2. UTE-MRI Imaging

The 3D-UTE-Cones MRI scans were performed on a 3T MRI scanner (MR750, GE Healthcare Technologies, WI, USA) using an eight-channel knee coil (one RF transmission channel and eight signal reception channels). The muscle-fat container was parallel to the B0 direction of the scanner, and all images were acquired in the coronal plane.

To measure T1 as a prerequisite for the two-pool UTE-MT modeling, the actual flip angle imaging-variable flip angle (AFI-VFA) sequence (AFI: TE = 0.032 ms, TRs = 20,100 ms, FA = 45°; VFA: TE = 0.032 ms, TR = 20 ms, FAs = 5°, 10°, 20°, and 30°) was performed [40]. Notably, the AFI-VFA sequence results in B1 corrected T1 measurement. A 3D-UTE-Cones-MT sequence (Fermi saturation pulse power = 500°, 1000°, and 1500°; frequency offset = 2, 5, 10, 20, and 50 kHz; FA = 7°; 11 spokes per MT preparation) was performed for MT ratio (MTR) measurement and two-pool MT modeling [34,41]. Field of view, in-plane matrix dimension, slice thickness, and total scan time were 12 cm, 192 × 192, 4 mm, and 35 min, respectively.

### 2.3. MRI Data Analysis

UTE-MRI analyses were performed initially within seven regions of interest (ROIs) covering only muscle to calculate T1, MTR, and MMF (from MT modeling). Each ROI was then gradually expanded by adding only the neighboring pure fat voxels (lard), which meant increasing FF within ROIs, while the originally included muscle voxels were kept intact. Finally, calculated T1, MTR, and MMFs were plotted against FF values to investigate their estimated variation levels when encountering fat infiltration in muscle. A 5% variation threshold from the original MRI measures (FF = 0) was considered as the range of robustness for each one of the MRI measures. Analyses were performed on a single slice in the middle of the phantom-fat phantom.

A single-component exponential model was used for T1 measurement. The acquired UTE-MT data with the three saturation pulse power levels and the five frequency offsets were used first for MTR calculation and then two-pool MT modeling to calculate MMF [33,34,41]. MMF calculation was repeated using (1) a constant T1 (T1 = 750 ms, i.e., average T1 values measured in pure muscle regions) and no B1 correction (i.e., B1 = 1), (2) a constant T1 and B1 correction (from the AFI sequence), (3) a measured T1 (from the AFI-VFA sequence) and no B1 correction, and (4) a measured T1 and B1 correction. It should be noted that the measured T1 values used as inputs in MT modeling were B1 corrected (from the AFI-VFA sequence). All UTE MRI measurements and models were performed using MATLAB (version 2021, The Mathworks Inc., Natick, MA, USA) codes developed in-house.

## 3. Results

Figure 1A demonstrates a T1-weighted UTE MRI image of the bovine muscle specimen embedded in lard (i.e., presumably FF = 100%). Figure 1B,C depict the T1 fitting curves (single component exponential model) on a variable FA dataset for a pure muscle and a pure fat ROI, respectively, both adequately far from the fat/muscle border to avoid partial-volume artifacts. 

Figure 2A shows an exemplary ROI highlighted in a blue box with an initial FF = 1 % that expanded to an ROI with FF = 70% by adding only the neighboring lard voxels (red box) while the muscle compartment of the expanding ROIs was intact. Figure 2B–D depict the UTE-T1 fittings within the three exemplary ROIs shown in Figure 2A (FF = 1, 35 and 70%). T1 values decreased by increasing FF (T1 = 676, 407, and 296 ms for FF = 1, 35 and 70%, respectively).

Figure 3 illustrates the two-pool UTE-MT modeling using a super-Lorentzian function over variable off-resonance frequencies for three different power levels (500°, 1000°, and 1500° power levels are shown in blue, green, and red lines) within four exemplary ROIs with (A1–A4) FF= 0 (pure muscle far from fat border, Figure 1A), (B1–B4) FF = 1% (Figure 2A), (C1–C4) 35% (Figure 2A), and (D1–D4) FF = 70% (Figure 2A). MT modeling was repeated four times using (1) a constant T1 (T1 = 750 ms) and no B1 correction, (2) a constant T1 and B1 correction, (3) a measured T1 and no B1 correction, and (4) a measured T1 and B1 correction. The actual signal changes within all investigated ROIs were fitted reasonably well using all the calculation methods. 

Figure 4 depicts T1 and MTR as well as their normalized values versus FF within the seven series of ROIs initiated with pure muscle ROIs with FF = 0% and expanded to ROIs up to FF = 70% by adding only neighboring pure lard voxels. T1 and MTR decreased by more than 5% for FF values above 10%. For ROIs with 50% fat content, T1 and MTR decreased to around 40% to 60 % of their original values.

Figure 5 shows MMF and normalized MMF values versus FF within the seven expanding ROIs (FF = 0 to 70%) using the four calculation methods used in Figure 3. MMF decreased by more than 5% for FF above 10% when constant T1 (T1 = 750 ms) was used (Figure 5A–D). However, when the measured T1 was used (E-H), the 5% MMF reduction occurred for FF > 30%. MMF calculated with the measured T1 showed a much more robust trend, particularly with a less than 3% error for FF < 20%. For ROIs with 50% fat content, the MMF error stayed below 30% when the accurately measured T1 was used in the calculation.

## 4. Discussion

The impact of fat content on the MMF estimations from UTE-MT modeling was investigated using a muscle/lard phantom. The estimated MMF from UTE-MT modeling using the accurately measured T1 demonstrated a robust trend and showed under 5% changes for muscle/fat ROIs with FF up to 30%. The MMF underestimation was less than 3% for regions with FF < 20%.

This study highlighted the potential of the two-pool UTE-MT modeling with the T1 compensation method for robust protein content estimation as a measure of muscle quality while remaining insensitive to fat infiltration up to moderate levels. This could be a particularly useful technique for some applications, such as analyzing rotator cuff musculature where pathologic muscles can contain up to 30% fat [39]. Notably, MMF estimation using constant T1 was robust only for regions with FF < 10%. The MTR and T1 values were also robust only for FF < 10%.

The presence of fat in muscle results in an apparent decrease in measured MT effect, which is demonstrated as decreases in MTR (Figure 4C,D) and MMF using constant T1 (Figure 5A,B). On the other hand, as shown in Appendix A, for a pure muscle ROI, decreasing the T1 value results in an overestimation of the MMF of muscle (increasing from around 9% to 24% by T1 decrease from 1000 ms to 300 ms). Such an overestimation in MMF was likely compensating for the apparent MT effect reductions caused by fat infiltration in our phantom study, which, in turn, led to the robust MMF values for a large range of FF (up to 30%) when the measured net T1 (including muscle and fat) was used in the models.

Robust estimation of MMF in muscle regardless of the macroscopic fat infiltration can decouple the muscle assessment from the fat infiltration, which may impair the muscle performance independently in the case of injuries and pathologies. Remarkably, UTE-MT modeling has been shown previously to be insensitive to tissue orientation and to provide a robust assessment of MSK tissues regardless of their orientation within the MRI scanner [37]. Being less sensitive to the tissue orientation and the macroscopic fat-filtration makes the UTE-MT technique a useful tool for muscle assessment, particularly in the myotendinous junction and regions with fibrotic tissues. It should be noted that UTE-MT modeling has been previously used mainly to assess MSK tissues with significant fractions of short T2 components, such as cartilage [42,43], meniscus [42], ligament [44], tendon [45,46,47], and bone [48,49,50,51,52].

MMF in muscle is potentially sensitive to the muscle quality and its ability to generate force. The fat present in muscle has always been a source of concern in MMF calculation. Li et al. have demonstrated significant underestimation in MMF values by increasing FF through a set of numerical simulations of MT pulses [16]. Therefore, using MT sequences combined with fat saturation methods has been suggested in previous studies to avoid MMF underestimations [11,15,16]. However, fat saturation preparation can create challenges. For instance, the fat saturation pulse may attenuate signals from tissues with short T2 relaxation times because of their broad frequency spectrum that may overlap with the main fat signal peak [53]. In addition, the pulse may generate magnetization transfer effects [54], which can result in increases in MMF measurement [11].

Several other quantitative MRI techniques have been reported in the literature for muscle assessments. T1 and T2 measurements have been used for muscle injury and pathology assessment and aging-related differences in several investigations [5,6,7,8,10]. Briefly, T1 values were found to be higher in muscle inflammation regions while they are lower in fat-infiltrated regions [5]. T2 values, however, were found to be higher in both muscle inflammation and fat-infiltrated regions [9]. DT imaging in MR has been widely used to evaluate the diffusion preference directions by the water molecules in tissues with well-organized long fibers, such as muscles, tendons, and ligaments. DT indices can provide information about the tissue microstructure and fibers’ orientation at a microscopic level. The diffusion properties of muscle, especially the third eigenvalue of the diffusion tensor and degree of diffusion anisotropy, reflect muscle damage due to injury [19,20] and disease [4,21]. Moreover, since macroscopic fat infiltration is a widespread observation in muscle pathologies and injuries, several studies have been devoted to performing an accurate estimation of the fat content in tissues, with most employing the different frequencies of fat and water protons [22,23,24,25,26,27,28,29,30].

The limitations of this study can be summarized in three aspects. First, the number of samples was small as is the nature of pilot studies. Second, bovine rather than human muscle tissue was used because bovine muscle could be readily obtained in a fresh and non-diseased state. Third, since the original muscle cuts were not performed by the authors, identifying the exact muscles of the bovine chuck (with more than 20 muscles) was not possible in this study. Considerable differences in fat and collagen composition among the muscles [55] and within each specific muscle [56] may affect the estimated MMF values. Although biochemical or microstructural analyses were not performed to confirm a minimum fat content in the muscle cuts, the selected ROIs were in regions that appeared devoid of macroscopic fat. Fourth, this study was performed ex vivo on a uniform muscle/fat phantom. The presence of nonuniform fat with irregular shapes and other soft and hard tissues, lower spatial resolution, a higher body temperature [57], and subject motion may all contribute to the reduced performance of all UTE-MRI-based imaging techniques in vivo compared with ex vivo studies. Fifth, this study was focused on macroscopic fat infiltration in muscle as limited by the current phantom study design. Future investigations should be performed to investigate the potential impacts of microscopic fat infiltration in muscle, which might require accurate and well-developed fat saturation or water excitation techniques [58].

## 5. Conclusions

Robust estimation of MMF in muscle regardless of the macroscopic fat infiltration can decouple the muscle assessment from the fat infiltration. The estimated MMF from UTE-MT modeling using the accurately measured T1 demonstrated a robust trend and showed under 5% changes for muscle/fat ROIs with FF up to 30%. MMF using a constant T1 value, similar to MTR showed robust values only for ROIs with FF < 10%. This study highlighted the potential of the UTE-MT modeling with the T1 compensation method for robust skeletal muscle assessment.

## Figures and Tables

**Figure 1 diagnostics-13-00876-f001:**
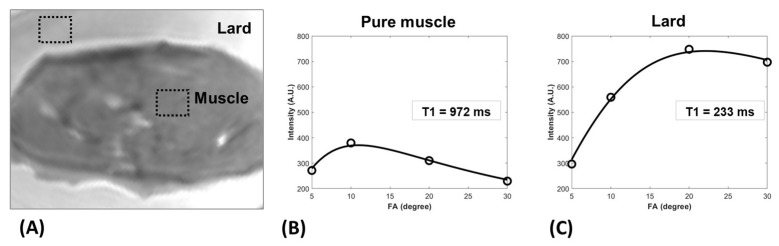
(**A**) T1-weighted UTE MRI image of a bovine muscle specimen embedded in lard (i.e., FF = 100%). UTE-T1 recovery fitting curves in (**B**) a pure muscle region of interest (ROI) (T1 = 972 ms) and a (**C**) pure fat ROI (T1 = 233 ms).

**Figure 2 diagnostics-13-00876-f002:**
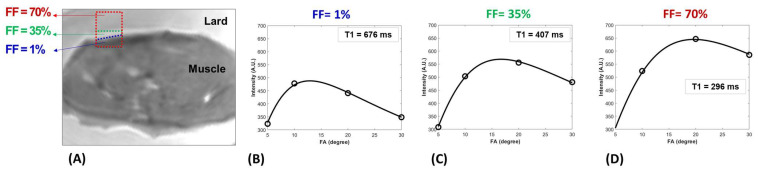
(**A**) T1-weighted UTE MRI image of the bovine muscle specimen embedded in lard. The blue box (dotted line) indicates an exemplary ROI with FF ≈ 1 % that expanded by only adding the neighboring lard voxels up to FF = 35% (green box, dotted line) and then FF = 70% (red box, dotted line). UTE-T1 recovery fitting curves for the ROI with (**B**) FF = 1% (T1 = 676 ms), (**C**) FF = 35% (T1 = 407 ms), and (**D**) FF = 70% (T1 = 296 ms).

**Figure 3 diagnostics-13-00876-f003:**
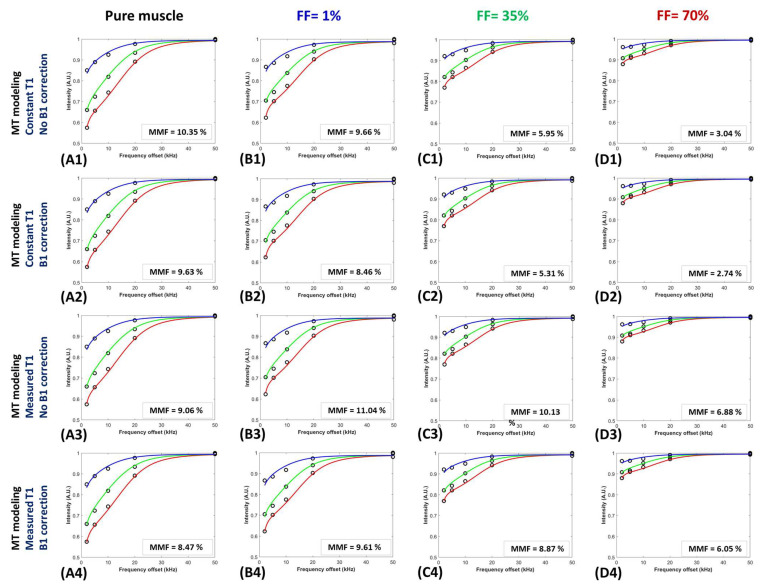
Two-pool UTE-MT modeling using a super-Lorentzian function over variable off-resonance frequencies for three different power levels (500°, 1000°, and 1500° power levels are shown in blue, green, and red lines) within four exemplary ROIs with (**A1**–**A4**) FF= 0 (pure muscle far from fat border, Figure 1A), (**B1**–**B4**) FF = 1% (Figure 2A), (**C1**–**C4**) 35% (Figure 2A), and (**D1**–**D4**) FF = 70% (Figure 2A). MT modeling was repeated using (1) a constant T1 (T1 = 750 ms, i.e., average T1 values measured in pure muscle regions) and no B1 correction (i.e., B1 = 1), (2) a constant T1 and B1 correction (from the AFI sequence), (3) a measured T1 (from the AFI-VFA sequence) and no B1 correction, and (4) a measured T1 and B1 correction.

**Figure 4 diagnostics-13-00876-f004:**
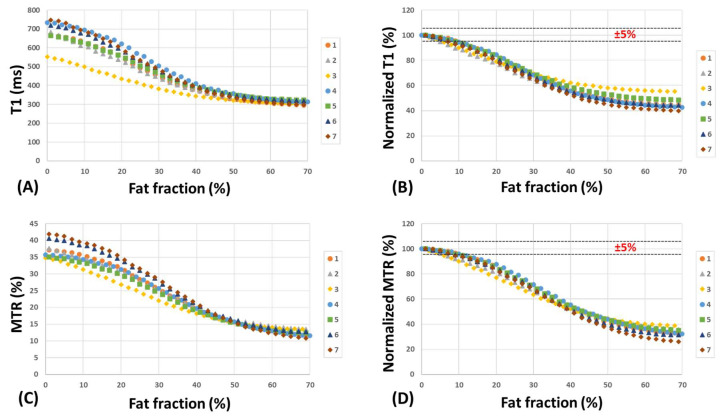
(**A**) T1 (ms) and (**B**) normalized T1 (%) versus FF within seven series of ROIs initiated with pure muscle ROIs (FF ≈ 0%) on the fat border and expanded by adding only neighboring pure lard voxels (FF up to 70%). (**C**) MTR (%) and (**D**) normalized MTR (%) versus FF within the same ROIs. T1 and MTR decreased by more than 5% for FF above 10%.

**Figure 5 diagnostics-13-00876-f005:**
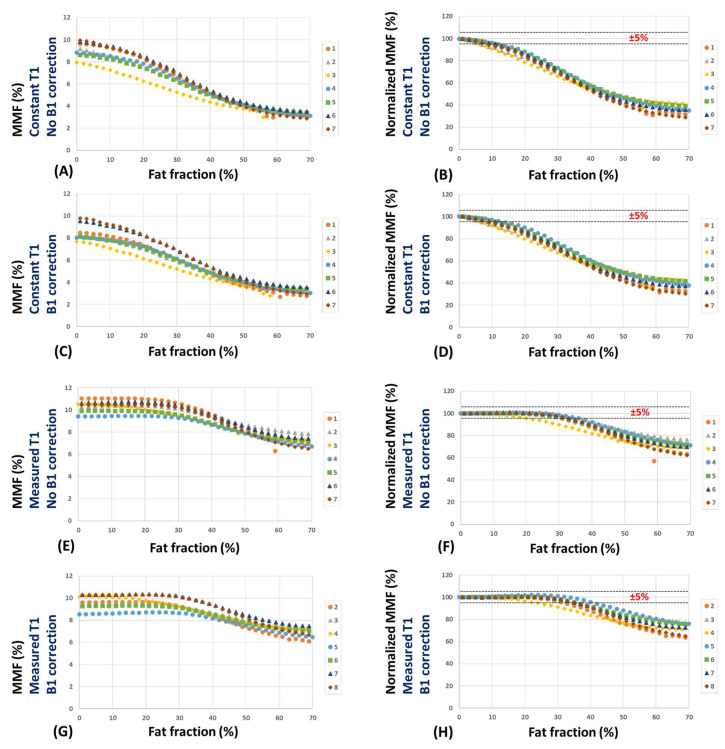
MMF and normalized MMF values versus FF within the seven expanding ROIs (FF = 0 to 70%) using (**A**,**B**) a constant T1 (T1 = 750 ms) and no B1 correction, (**C**,**D**) a constant T1 and B1 correction, (**E**,**F**) a measured T1 (from the AFI-VFA sequence) and no B1 correction, and (**G**,**H**) a measured T1 and B1 correction. MMF decreased by more than 5% for FF above 10% when constant T1 (T1 = 750 ms) was used (**A**–**D**). However, when an accurately measured T1 was used (**E**–**H**), the 5% MMF reduction occurred for FF > 30%. The calculated MMF using a measured T1 showed a less than 3% error for FF < 20%.

## Data Availability

Data supporting the reported results can be provided by the corresponding authors upon courtesy request.

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
