# Peer review of "Robust Assessment of Macromolecular Fraction (MMF) in Muscle with Differing Fat Fraction Using Ultrashort Echo Time (UTE) Magnetization Transfer Modeling with Measured T1"

_diagnostics, 2023, doi:10.3390/diagnostics13050876_

Round 1

Reviewer 1 Report

The experimental study is original and well carried out, although preparatory to further investigations necessary in the future. It opens up various possibilities for in-depth study not only in animal models to solve some critical issues analyzed in the work and define the lines of development for a possible practical application in both human and veterinary medicine.

1. This study highlighted the potential of the UTE-MT modeling with accurate T1 measurement for robust muscle assessment.

2. Yes, I consider the topic original. The study is preparatory, it opens up interesting scenarios but needs further insights.

3. In my opinion, the document is well structured and balanced in its current form. This is a  preparatory study which establishes important elements for future studies.

 4. The conclusions are consistent with the evidence and arguments presented. The paper is preparatory to further studies.

 5. The references are appropriate.

Author Response

We highly appreciate your support.

Reviewer 2 Report

The work is well structured, where a conspicuous and well-valued scientific basis emerges.

The use of Nuclear Magnetic Resonance with the Imaging modality, as an innovative and structural equipment, makes the manuscript of high scientific value combined with the reported results.

Author Response

We highly appreciate your support.

Reviewer 3 Report

In this paper, the authors present a new technique to determine the macromolecular fraction parameter in muscle regardless of the macroscopic fat infiltration. Different measurements were obtained from specific regions of a phantom prepared with muscle and lar. This technique could help to improve the muscle assessment, traditionally conducted with magnetic resonance imaging. The paper is well written and structured and the conclusions are supported by the results. As a general comment I suggest to the authors to provide more specific information about the type of muscle (architecture, composition, etc. ) used in the preparation.

Specific comments

Line 69: MSK means musculoskeletal?

Line 86: Which muscles were selected?

Line 88: What were the dimensions of the container?

Line 147: Why 0% here for the fat fraction in the blue box while in the text and in the figure title is specified as 1%? 

Line 214: A different text font appears in this sentence.

Line 233: “… few studies have reported significant underestimation of MMF for muscles with a considerable fat infiltration [16]” This idea has already been used in the introduction section. I’d suggest to be more specific here in this section. 

Line 269: This sentence should be rewritten according to the findings of the work to avoid misunderstandings. Muscle impairment influenced by fat infiltration has not been assessed in this study.

Author Response

C: Reviewer #3:

In this paper, the authors present a new technique to determine the macromolecular fraction parameter in muscle regardless of the macroscopic fat infiltration. Different measurements were obtained from specific regions of a phantom prepared with muscle and lar. This technique could help to improve the muscle assessment, traditionally conducted with magnetic resonance imaging. The paper is well written and structured and the conclusions are supported by the results.

Response: We appreciate your support, valuable comments, and time. Please note that the changes in the manuscript are highlighted in yellow and labeled with the corresponding comments (e.g., R3.1 refers to comment No. 1 from the third Reviewer).

  1. As a general comment I suggest to the authors to provide more specific information about the type of muscle (architecture, composition, etc. ) used in the preparation.

Response: We appreciate the suggestion. Fresh chuck cuts of lean bovine muscles were purchased from a local grocery store. Since the original muscle cuts were not performed by the authors, identifying the exact muscles of the bovine chuck (with more than 20 muscles [62]) was not possible in this study. We acknowledge that the considerable differences in fat and collagen composition between the muscles [62] and within each specific muscle [63] may affect the estimated MMF values. Although biochemical or microstructural analyses were not performed to confirm a minimum fat content in the muscle cuts, the selected ROIs were in regions that appeared devoid of macroscopic fat. This information and the limitation have been added to the Methods and Discussion sections.

  1. Seggern, D.D.V.; Calkins, C.R.; Johnson, D.D.; Brickler, J.E.; Gwartney, B.L. Muscle Profiling: Characterizing the Muscles of the Beef Chuck and Round. In Proceedings of the Meat Science; September 2005; Vol. 71, pp. 39–51.
  2. Picard, B.; Gagaoua, M.; Gagaoua, M. Muscle Fiber Properties in Cattle and Their Relationships with Meat Qualities: An Overview. J Agric Food Chem 2020, 68, 6021–6039.

Specific comments

  1. Line 69: MSK means musculoskeletal.

Response: That is correct. The MSK definition has been added in the new version of the manuscript before using its abbreviation. 

  1. Line 86: Which muscles were selected?

Response: We appreciate the comment. The muscle was from a bovine chuck cut containing muscles from the upper shoulder. This information has been added to the Methods as follows:

Fresh cuts of lean bovine chuck muscles and pure pork fat (lard) were purchased from a local grocery store.

Unfortunately, the exact muscle used in the bovine thigh has not been recorded.

  1. Line 88: What were the dimensions of the container?

 Response: We appreciate the comment. The diameter and length of the container were 10 and 20 cm, respectively. This information has been added to the manuscript as follows:

Two muscle sections were placed in a cylindrical plastic container (10 cm diameter and 20 cm length, approximately).

  1. Line 147: Why 0% here for the fat fraction in the blue box while in the text and in the figure title is specified as 1%? 

Response: We appreciate the comment. The typo in the caption of Figure 2 has been revised to FF ≈ 1%.

  1. Line 214: A different text font appears in this sentence.

Response: We appreciate the comment. The font in this paragraph has been matched with the rest of the manuscript.

  1. Line 233: “… few studies have reported significant underestimation of MMF for muscles with a considerable fat infiltration [16]” This idea has already been used in the introduction section. I’d suggest to be more specific here in this section.

Response: We appreciate the comment. The sentence has been revised as follows:

MMF in muscle is potentially sensitive to the muscle quality and its ability to generate force. The fat present in muscle has always been a source of concern in MMF calculation. Li et al. have demonstrated significant underestimation in MMF values by increasing FF through a set of numerical simulations of MT pulses [16].

  1. Line 269: This sentence should be rewritten according to the findings of the work to avoid misunderstandings. Muscle impairment influenced by fat infiltration has not been assessed in this study.

Response: We appreciate the comment. To avoid misunderstanding, the last section of the sentence has been deleted as follows:

Robust estimation of MMF in muscle regardless of the macroscopic fat infiltration can decouple the muscle assessment from the fat infiltration, which may impair the muscle performance independently in the case of injuries and pathologies